The periodical cicada four-year acceleration hypothesis revisited and the polyphyletic nature of Brood V, including an updated crowd-source enhanced map (Hemiptera: Cicadidae: Magicicada)

Cooley John R. john.cooley@uconn.edu jcooley@wesleyan.edu 1 2
Arguedas Nidia 3
Bonaros Elias 4
Bunker Gerry 5
Chiswell Stephen M. 6
DeGiovine Annette 4
Edwards Marten 7
Hassanieh Diane 2
Haji Diler 2
Knox John 8
Kritsky Gene 9
Mills Carolyn 10
Mozgai Dan 11
Troutman Roy 12
Zyla John 13
Hasegawa Hiroki 14
Sota Teiji 14
Yoshimura Jin 15
Simon Chris 2
1 College of Integrative Sciences, Wesleyan University , Middletown , CT , United States of America
2 Department of Ecology and Evolutionary Biology, The University of Connecticut , Storrs , CT , United States of America
3 Cleveland Metroparks , Cleveland , OH , United States of America
4 Long Island , NY , United States of America
5 Massachusetts Cicadas , Marlborough , MA , United States of America
6 National Institute of Water and Atmospheric Research , Wellington , New Zealand
7 Department of Biology, Muhlenberg College , Allentown , PA , United States of America
8 Department of Biology, Washington and Lee University , Lexington , VA , United States of America
9 Department of Biology, Mount St. Joseph University , Cincinnati , OH , United States of America
10 Research Services, The University of Connecticut Libraries , Storrs , CT , United States of America
11 Cicada mania , NJ , United States of America
12 Batavia , OH , United States of America
13 Mid-Atlantic Cicadas, MD , United States of America
14 Department of Zoology, Graduate School of Science, Kyoto University , Kyoto , Japan
15 Graduate School of Science and Technology, Shizuoka University , Hamamatsu , Japan
Wallis Graham
Electronic publication date: 2018 Jul 31
Publication date: 2018
Volume: 6
Electronic Location ID: e5282
Received 2018 Mar 4; Accepted 2018 Jun 26
Copyright: ©2018 Cooley et al.
Copyright year: 2018
Copyright holder: Cooley et al.
License: This is an open access article distributed under the terms of the Creative Commons Attribution License, which permits unrestricted use, distribution, reproduction and adaptation in any medium and for any purpose provided that it is properly attributed. For attribution, the original author(s), title, publication source (PeerJ) and either DOI or URL of the article must be cited.
License URL: https://creativecommons.org/licenses/by/4.0/

Keywords: Periodical Cicada, Distribution, Mapping, Climate, Biogeography, mtDNA, Crowdsourcing

Funding: National Geographic CRE National Science Foundation NSF DEB 04-22386 DEB-09-55849 DEB 16-55891 University of Connecticut JSPS KAKENHI JP22255004 JP22370010 JP26257405 JP15H04420 Shizuoka University The National Geographic CRE-sponsored project “Making Modern Maps of Magicicada Emergences” provided funding for this project. This work is partially supported by the National Science Foundation under Grant Nos. NSF DEB 04-22386, DEB-09-55849, DEB 16-55891 to Chris Simon and J. Cooley (and others) and grants from the University of Connecticut, Vice President’s Research Excellence Program. This study was partly supported by JSPS KAKENHI Grant Numbers JP22255004, JP22370010, JP26257405, and JP15H04420 to Jin Yoshimura. We also received funding from Shizuoka University. There was no additional external funding received for this study. The funders had no role in study design, data collection and analysis, decision to publish, or preparation of the manuscript.

==============================
The periodical cicadas of North America (Magicicada spp.) are well-known for their long life cycles of 13 and 17 years and their mass synchronized emergences. Although periodical cicada life cycles are relatively strict, the biogeographic patterns of periodical cicada broods, or year-classes, indicate that they must undergo some degree of life cycle switching. We present a new map of periodical cicada Brood V, which emerged in 2016, and demonstrate that it consists of at least four distinct parts that span an area in the United States stretching from Ohio to Long Island. We discuss mtDNA haplotype variation in this brood in relation to other periodical cicada broods, noting that different parts of this brood appear to have different origins. We use this information to refine a hypothesis for the formation of periodical cicada broods by 1- and 4-year life cycle jumps.

Introduction

North American periodical cicada (Magicicada) adults emerge synchronously, predictably, and in overwhelming numbers; their black bodies, orange wing veins, striking red eyes, and loud acoustical choruses are unmistakable. Although early European settlers in North America mistook these insects for migratory locusts (Kritsky, 2004), their sudden appearances and equally sudden disappearances are not caused by movement, but by long life cycles (13- or 17- years) spent mostly underground with only a brief aboveground adult phase. Perhaps one of the strangest aspects of Magicicada is the existence of broods, which are multispecies assemblages in which all members, regardless of species, emerge synchronously on the same schedule. Across the eastern and central United States, the broods form a patchwork pattern of interlocking year classes that are generally parapatric, temporally offset, and resemble puzzle pieces. Such spatial relationships suggest the existence of some kind of competitive interactions that limit brood overlaps, such as underground competition among nymphs and/or aboveground competition among adults (Bulmer, 1977; Lehmann-Zeibarth et al., 2005). These spatial patterns also suggest some mechanism for brood formation by which broods give rise to each other and thus have parent–child relationships (Marlatt, 1902). While Magicicada are well-known for their fixed life cycles, two kinds of life cycle shifts have been demonstrated; permanent life cycle shifts (e.g., cicada populations change cycles and stay on their new cycle; Martin & Simon, 1988; Martin & Simon, 1990; Marshall & Cooley, 2000; Simon et al., 2000; Cooley et al., 2001), and temporary life-cycle shifts (e.g., cicada populations adopt the alternate cycle but return to their original cycle; Simon & Lloyd, 1982; Kritsky, 1988; Marshall, Cooley & Hill, 2011; Marshall, Hill & Cooley, 2018). While both kinds of life-cycle shifts were hypothesized by Lloyd & Dybas (1966), temporary life-cycle switching is thought to be responsible for brood formation within life-cycles (Marlatt, 1902; Lloyd & Dybas, 1966; Lloyd & White, 1976; Simon & Lloyd, 1982).

Given the spatial and temporal relationships of the 17-year broods, Lloyd & Dybas (1966) developed a general “4-year acceleration” hypothesis by which most broods could be derived by simple 1- or 4-year temporary life cycle advancements (or “accelerations”) from an ancestral parent brood on the same schedule as Brood XIV (Fig. 1; see also Lloyd & White, 1976; Simon, 1983). The discovery of persistent disjunct “miniature” brood isolates on Long Island, New York and elsewhere (Simon & Lloyd, 1982; Cooley, 2015; Cooley et al., 2015) and genetic evidence that many broods have complex, multiple origins (Martin & Simon, 1988; Martin & Simon, 1990; Simon et al., 2000; Cooley et al., 2001; Sota et al., 2013) suggest that the process modeled by this hypothesis, in which populations become temporally isolated and form new broods or join existing ones, may be more common than previously thought and apply to more than just the known disjunct populations. Here we test whether the main body of Brood V has a single origin, or whether it reflects multiple episodes of brood formation.

Figure 1 The original Lloyd and Dybas “4-year acceleration” scheme (Lloyd & Dybas, 1966).

Broods in parentheses were considered of uncertain existence.

Seventeen-year periodical cicada Brood V, whose range includes a variety of different climate and habitat types, emerged in 1931, 1948, 1965, 1982, 1999, and 2016. Brood V contains all three named 17-year cicada species (M. septendecim, M. cassini, and M. septendecula) and has a reported range whose various parts are in close proximity to or in contact with Broods I, VIII, IX, X, and XIV (Marlatt, 1923; Simon & Lloyd, 1982; Simon, 1988). Brood V is close to the boundary between eastern and middle periodical cicada haplotypes as defined by Sota et al. (2013), but sampling in that paper did not encompass the full range of Brood V. In this paper, we genotype samples of Brood V and neighboring Brood I in Virginia and West Virginia, collate historical records, and create a new georeferenced map of Brood V’s 2016 emergence. Our genetic, geographic, and historical data suggest that like Broods IX and X (Sota et al., 2013), Brood V is composed of at least four subpopulations with separate evolutionary origins and that the main range of Brood V is not of a single origin but rather consists of fused eastern and western populations. We propose hypotheses for the formation of these subpopulations and review the temporal and geographic relationships of other broods in contact with Brood V. We suggest that Lloyd & Dybas’s (1966) “4-year acceleration” hypothesis be expanded to consider decelerations (or temporary life cycle retardations), which are sometimes more geographically parsimonious. We also document changes in the distribution of Brood V in parts of its range.

Methods

Verified records

In May and June 2016, we collected records of the Brood V emergence across its range by searching for active choruses on days with appropriate weather conditions. We used mobile GPS dataloggers to collect information about the density of cicadas and the species present (more details of the methods can be found in Cooley et al., 2013; Cooley et al., 2016). In northeast Ohio, particularly Cuyahoga, Medina and Lake counties, and in eastern Suffolk County (Long Island) New York, data were collected by trained surveyors with handheld GPS devices visiting preselected locations on multiple days throughout the emergence. Distributional information was organized and mapped using ArcGIS 10.5 (ESRI, 2017). Cuyahoga and Medina county Ohio data were deposited in a database managed by Cleveland Metroparks. All data were also included in the periodical cicada database found at http://www.magicicada.org.

Crowdsourced records

The website http://www.magicicada.org and Cleveland Metroparks collected unverified (“crowdsourced”) periodical cicada sightings from the general public in May 2016. We used these records to inform our decisions about areas to map in detail. Individual crowdsourced records are not necessarily reliable; thus, we weighted crowdsourced records by assigning higher confidence to records that were clustered or that were in close proximity to verified records as described in Cooley et al. (2015). After we had stopped collecting data from the general public, we received an unusual number of reports from Carbon County, Pennsylvania, via the website http://www.cicadamania.com/ which we then investigated.

Overlaps

Because roads are sparse in the parts of Virginia where Brood V contacts Brood I, direct measurements of overlaps are impractical. Instead, we made conservative estimates of overlaps by constructing linear features in Arc GIS 10.5 that linked verified presence observations of Brood I with verified presence records of Brood V. To do so we looked for instances in which series of verified records of one brood are clearly within territory occupied by the other brood; for example, if we collected 11 records along a route occupied by Broods I and V in the order (V1 V2 V3 I4 V5 I6 V7 I8 I9 I10 I11), then we constructed a single linear feature connecting (I4 V5 I6 V7). We extracted the lengths of all linear features as conservative estimates of overlap. We did not use records of single individuals as endpoints of these features.

Historical data

Historical data from Brood V can be found in Marlatt (1923), Simon (1988), Kritsky, Smith & Gallagher (1999; for Ohio only), and Simon & Lloyd (1982; for Long Island only). Additional historical records were gleaned from Magicicada published literature, museum specimens in the University of Michigan Museum of Zoology (UMMZ) and from unpublished field notes and records in the http://www.magicicada.org database. To verify the historical presence of the newly discovered disjunct population in Carbon County, Pennsylvania, we gathered data from archived copies of newspapers in the Carbon County area for the Brood V emergence years 1883, 1897, 1931, 1948 and 1965 and 1999. We searched for the terms “cicada” or “locust” (an archaic term for cicadas used in North America (Kritsky, 2001; Kritsky, 2004) in the months of January through August in three newspaper archives: Library of Congress’s Chronicling America Historic American Newspapers, the Pennsylvania Historical Newspapers, and http://Newspapers.com.

Genetic data

We amplified and sequenced a 525-bp portion of mitochondrial cytochrome oxidase subunit I (COI) for 64 M. septendecim specimens collected in 2016 from Brood V, and in 2012 from Brood I using methods described in Sota et al. (2013). Haplotypes were compared to previously reported mitochondrial haplotypes of M. septendecim from Broods I and V (Sota et al., 2013) to determine haplotype groups. We used TCS version 1.21 to construct a haplotype network (Clement, Posada & Crandall, 2000). Details of specimens and accession numbers of the reference haplotype sequences are given in Table 1. Methods are identical to those in Sota et al. (2013).

Table 1 Mitochondrial haplotypes of select Brood V and Brood I M. septendecim specimens.

See Fig. 7 for mapped locations of specimens.

Brood	Year	Location	State	County	Latitude	Longitude	No. of samples	Sample code (haplotype group: haplotype accession)	Haplotype group	
I	1978	A	NY	Suffolk	40.903	−72.862	3	en0001 (Ae: AB740808) en0002 (Ae: AB740809) en0005 (Ae: AB740810)	Ae	
I	2012	B	VA	Shenandoah	38.6861	−78.7338	4	V68B	Ae	
I	2012	C	WV	Pendleton	38.515	−79.1404	4	en1027 (Am); en1028 (Am); en1029 (Am: AB740828); en1030 (Ae: AB740808)	Ae, Am	
I	2012	D	WV	Pendleton	38.5121	−79.2557	4	en1023,1024,1026 (Am: AB740828); en1025 (Am: AB74026)	Am	
I	2012	E	VA	Rockingham	38.3853	−79.0246	3	en1013-1015 (Ae: AB740808)	Ae	
I	2012	F	WV	Pendleton	38.3041	−79.1918	4	en0913,0914,0916 (Ae: AB740808); en0915 (Ae)	Ae	
I	2012	G	VA	Augusta	38.2152	−79.1232	4	en1016-1019 (Ae: AB740808)	Ae	
I	2012	H	VA	Augusta	38.1614	−79.1741	3	en1020-1022 (Ae: AB740808)	Ae	
I	2012	I	VA	Rockbridge	37.84432	−79.59543	4	12.VA.RKB (Ae: AB740808)	Ae	
I	2012	J	VA	Botetourt	37.5555	−79.6355	4	V57	Ae	
V	1999	K	OH	Ross	39.317	−82.777	3	en0011 (Am: AB740826) en0012 (Am: AB740827) en0013 (Am: AB740828)	Am	
V	2016	L	VA	Shenandoah	38.87299	−78.63828	4	en0961-0964 (Ae: AB740808)	Ae	
V	2016	M	VA	Augusta	38.21769	−79.21769	4	en1001-1004 (Ae: AB740808)	Ae	
V	2016	N	VA	Augusta	38.1839	−79.21317	4	en0957-0960 (Ae: AB740808)	Ae	
V	2016	O	VA	Bath	38.0921	−79.78225	4	en1009-1012 (Am: AB740828)	Am	
V	2016	P	VA	Bath	37.95093	−79.86982	4	en1005,1007 (Am: AB740828); en1006 (Ae: AB740808); en1008 (Am)	Ae, Am	
V	2016	Q	VA	Alleghany	37.85146	−79.80739	4	en0997-0999 (Am: AB740828); en1000 (Ae: AB740808)	Ae, Am	

Results

Brood V consists of four distinct parts: (1) a large body of the brood extending from the Ohio Valley to Lake Erie; (2) a nearby but separate, southeastern portion associated with the Shenandoah Valley and in close contact with Brood I; (3) a small disjunct population of M. septendecim in Carbon County, Pennsylvania, associated with the Lehigh River in and around the town of Jim Thorpe; and (4) a previously known disjunct population of M. septendecim on eastern Long Island containing only M. septendecim (Figs. 2–5). We found that Brood V in Virginia is largely parapatric with Brood I; for the five Virginia locations where we found measurable overlap, estimated overlaps were limited to 0.51 ± 0.45 km, with the largest overlap less than 1.3 km (Fig. 6).

Figure 2 Overview of 17-year periodical cicada Brood V records from 2016.

Orange circles are positive records, while gray circles are negative (absence) records. Symbol size reflects weights applied to reflect confidence (see text). Smaller symbols are crowdsourced records with low confidence; larger symbols have higher confidence (note that in Southern VA crowdsourced records in the range of Brood II were checked by the authors a week or more after the reports; we found no evidence of cicadas and their absence is indicated by grey dots).

Figure 3 Ohio Valley and Virginia portions of 17-year periodical cicada Brood V.

Orange symbols are verified positive records; gray symbols are verified 2016 negative records. Circles are presence records from 2016, squares from 1999, and crosses from 1982. Symbol size reflects weights applied to reflect confidence (see text). Smaller symbols are crowdsourced records with low confidence; larger symbols have higher confidence.

Figure 4 Long Island portion of 17-year periodical cicada Brood V.

Orange symbols are verified positive records, gray symbols are verified negative records. Circles are records from 2016, squares from 1999, and crosses from 1982. Green symbols are Brood XIV records, and the single purple symbol is a Brood I record from 1978. Protected areas are shown in light green.

Figure 5 Carbon County, PA portion of 17-year periodical cicada Brood V as mapped in 2016.

Orange symbols are verified positive records; gray symbols are verified negative records. Brood II presence records are shown in red and reprinted from Cooley et al. (2015).

Figure 6 Associations of 17-year periodical cicada Broods.

Periodical cicada Broods V (orange symbols), I (purple symbols; Cooley, 2015), II (red symbols Cooley et al., 2015), IX (pink symbols) and X (blue symbols; Cooley et al., 2009).

Crowdsourced records

In this study, we evaluated 1,361 crowdsourced data points, most of which were within the range of Brood V, but some of which were from as far away as California (well outside the range of Magicicada). Crowdsourced records suggest that we did not miss any significant populations (Figs. 2– 4). The existence of the Carbon County, Pennsylvania population was initially revealed to us by records provided by the general public.

Genetic data

We obtained COI sequences for 64 M. septendecim from 17 populations in Ohio, West Virginia, and Virginia (Table 1). All haplotypes are variants of mitochondrial lineage A, which is restricted to M. septendecim and M. neotredecim (Martin & Simon, 1988; Martin & Simon, 1990; Marshall & Cooley, 2000; Simon et al., 2000; Cooley et al., 2001; Sota et al., 2013). Midwestern portions of Brood V contain only the Midwest haplotype, Am. Eastern populations of Brood V possess the Eastern haplotype, Ae. Brood V populations in mountainous areas along the Virginia-West Virginia border contain both haplotypes Ae and Am in the same forest patches in multiple locations (Fig. 7). Disjunct populations of Brood V on Long Island were entirely Ae. Populations from Jim Thorpe, Pennsylvania were not genotyped.

Figure 7 Periodical cicada mtDNA haplotypes in Broods I and V.

Haplotype Ae (triangles) and Am (circles) are found in 17-year periodical cicada Broods V (orange symbols) and I (purple symbols) following terminology of Sota et al. (2013). Genetic data from Table 1; haplotype network constructed using TCS version 1.21 (Clement, Posada & Crandall, 2000).

Historical Records and Population Status

Changes in brood ranges and abundance in Ohio

The recession of Brood V boundaries in Ohio has been discussed for over a century. Webster mapped the 1897 emergence, noting that in Erie County, periodical cicadas could be found in an area that was “little more than a peninsula-like extension, and will probably not appear again.” Gossard (1916) surveyed the next emergence in Erie County and documented that the brood persisted for another generation. Forsythe (1976) checked the distribution in 1965 and found cicadas just south of the Erie County line. The brood did emerge in 2016 just north of the county line, suggesting that there may have been doubt about the line when mapping the 1965 and later emergences.

The recession of the southwestern Ohio limits of Brood V, noted by Kritsky, Smith & Gallagher (1999), continued with the 2016 emergence. The emergences in Ross, Pike, and Scioto counties were scattered, with only light chorusing observed in the eastern portion of the counties compared to strong choruses reported in 1897. Forsythe (1976) noted that such light emergences suggest that “relatively small, isolated locations may be more persistent than originally supposed.”

Carbon County Pennsylvania disjunct population

Even though the Carbon County Pennsylvania disjunct of Brood V (Fig. 5) has not been reported in the scientific literature until now, it has been documented in museum collections and local newspaper reports. We found 12 M. septendecim in the UMMZ collected 6∕23∕1982 by J. A. Lankalis in Mauch Chunk (prior to 1953, Jim Thorpe was named Mauch Chunk, and the name has persisted because it is also the name of a nearby ridge). The UMMZ collection also contains four M. septendecim collected 5∕13∕1982 in Lehigh Gorge. Jim Thorpe is the gateway to the Lehigh Gorge.

The earliest news report of periodical cicadas on the same schedule as Brood V in Carbon County Pennsylvania is from 1880; in March of that year, the Carbon Advocate, of Leighton, predicted that periodical cicadas would appear that summer “in accordance with long established customs;” the same paper later reported on May 29 that a substantial emergence had occurred. In June 1897, the Allentown Morning Call noted the emergence of periodical cicadas in Schuylkill County, which is immediately to the southwest of Carbon County. While we did not find any periodical cicadas in Schuylkill County in 2016, we did not search the county exhaustively, and we did find periodical cicadas in Carbon County within 7 km of the Carbon/Schuylkill County line. In June 1931, the same newspaper reported periodical cicadas in Leighton and Mauch Chunk. The Allentown Morning Call also reported periodical cicadas in June 1948 and June 1965, with specific mentions of Mauch Chunk and Carbon County. In sum, these newspaper reports seem largely credible, since they describe cicadas in approximately the correct locations in Brood V emergence years; thus, these records support the existence of this disjunct at least as far back as 1880. Complete details of Pennsylvania Brood V historical records are included in the (Data S1).

Long Island disjunct populations

Brood XIV is the main brood on Long Island and only small pockets of Broods I, V, and IX exist in a restricted area of northeastern Long Island (Simon & Lloyd, 1982). The Long Island disjunct of Brood V (Fig. 4) was first recorded in 1914 by WT Davis (Simon & Lloyd, 1982). Like the Long Island disjuncts of Broods I, IX and X (Simon, Karban & Lloyd, 1981; Simon & Lloyd, 1982; Cooley et al., 2009; C. Simon field notes), these populations of Brood V appear to be declining and may go extinct in the near future. For example, Long Island Brood X has declined precipitously since 1970; an article in Newsday (Nelson, 2004) documented only small numbers of Brood X individuals emerging on Long Island in 2004 in areas that had large emergences in 1987 (C. Simon field notes) and 1970, including protected areas such as Connetquot State Park. Similarly, detailed qualitative records of Brood V in Wildwood State Park, Wading River, Long Island in 1982 and 1999, (C. Simon field notes; Magicicada database) and our survey in 2016, and anecdotal reports from residents of Calverton suggest a steep decline in Magicicada density both in protected and developed areas of north eastern Long Island. No records of Brood XI on Long Island exist beyond those recorded by W.T. Davis in the early 1900’s and reported in Simon & Lloyd (1982).

Discussion

Crowdsourcing as a biological tool

Crowdsourcing has been a key component of periodical cicada mapping ever since C. L. Marlatt constructed maps by enhancing older datasets with reports from state entomologists, agricultural experiment station bulletins, weather service observers, post offices, and entomological enthusiasts (Marlatt, 1898). Our crowdsourced periodical cicada records continue the tradition of citizen science successfully contributing to this effort. The emergence and evolution of citizen science as a subdiscipline of ecology and conservation biology, along with its challenges and opportunities have been reviewed by Silvertown (2009) and Pocock et al. (2017b). Recently, citizen science has emerged as a tool for evaluating species responses to forest management plans (Mair & Ruete, 2016; Mair et al., 2017), mapping the arrival and spread of invasive species (Pocock et al., 2017a), pollinator monitoring (Roy et al., 2016), and tracking monarch butterfly populations (Schultz et al., 2017). The citizen science approach is particularly powerful in the context of periodical cicada research due to the large range of potential habitat, the short duration of adult emergences, and the limited number of “expert” mappers. Notably, the Carbon County Pennsylvania Brood V disjunct was originally brought to our attention on the basis of citizen science reports; given the small size of this population, it is unlikely that we would have found it otherwise. We expect that citizen science efforts will continue to play an important role in monitoring whether the reported ranges of periodical cicada broods remain stable or undergo changes in the future.

Brood V has multiple origins

Although some morphometric studies suggested that broods are single-origin monophyletic assemblages (Simon, 1983), Marlatt (1902) acknowledged the possibility of broods having multiple origins, noting that, “In the case of a widely-scattered brood ... it is quite possible that certain swarms originated from a later-appearing brood by retardation of individuals, and other swarms from an earlier brood by acceleration in time of appearance of individuals ... but with the broods presenting a compact range a singleness of origin is evident” (Marlatt, 1902). Marlatt considered Brood V to have a “compact range.” He was unaware of the small disjunct populations of Brood V in Carbon County Pennsylvania and on Long Island, which by his reasoning, would be best explained as independently derived from a different brood.

Genetic data also indicate that the different parts of Brood V have unique histories, extending the findings of Sota et al. (2013) that Broods VI, IX, X, and XIV have multiple origins. Brood V M. septendecim from Midwestern localities belong to a central US mtDNA clade (Am) shared with all other Midwestern M. septendecim brood populations, while Brood V M. septendecim from eastern portions of the range belong to an eastern mtDNA clade (Ae), shared with all other eastern M. septendecim brood populations (Sota et al., 2013). In the middle of the main body of Brood V, in the Shenandoah Valley of Virginia and surrounding areas extending into West Virginia, there is a boundary between haplotypes Ae and Am. In this region, Brood V overlaps Brood I, which also contains both haplotypes Ae and Am, as does Brood IX (Table 1, Fig. 7; Sota et al., 2013). The spatial, temporal, and genetic relationships of Broods I, V, and IX suggest that they are derived from each other by 4-year jumps (discussed below), and because Brood V has the westernmost extent of these broods, it may have been a conduit for Midwestern haplotype Am to enter eastern populations of Broods I and IX. Further exploration of this hypothesis awaits detailed mapping of Brood IX in 2020 and more fine-scale genetic data for both mtDNA and nuclear genomes.

Four-year jumps

Magicicada stragglers, or off-cycle cicadas, are a well-documented phenomenon that has complicated efforts to map broods (Marshall, 2001). If stragglers emerge in sufficient numbers to satiate predators, then they may establish populations on a new schedule. Early emergences, or accelerations (written as “−4”), have been suggested to result from crowding or better than average environmental conditions (e.g., longer growing seasons) that allow a fraction of the population to complete development ahead of schedule (Williams & Simon, 1995). Unexpectedly late emergences, or decelerations (written as “ +4”), may be caused by worse than average environmental conditions that prevent a fraction of the population from completing development in the expected time (see Karban, Black & Weinbaum, 2000; also Lloyd & Dybas, 1966; Lloyd & White, 1976). The spatial association of brood pairs (Alexander & Moore, 1962) offset by one year (I/II, III/IV, IX/X, XXII/XXIII) and by 4 years (XIV/I/V/IX, II/VI/X/XIV, XIX/XXIII) suggests that both 1- and 4-year jumps are important for shaping Magicicada spatiotemporal patterns (Lloyd & Dybas, 1966). Among the possible jumps that have been documented (Cooley et al., 2011; Marshall, Cooley & Hill, 2011; Cooley et al., 2016), 4-year jumps appear particularly common; Marshall, Hill & Cooley (2018) used repeated censuses to demonstrate that individual cicadas are more likely to accelerate by 4 years than by 1 year. Furthermore, no straggler emergences dense enough to form choruses have been documented with 1-year offsets from their broods, while straggler choruses with 4-year offsets are common (see below). Whether 4-year jumps are favored by some unknown selective advantage, or whether they are a reflection of the developmental processes underlying periodical cicada life cycles remains unknown.

Four-year accelerations are well-documented in Magicicada. We hypothesize that accelerations occur during warmer times when growing seasons are lengthened and cicadas can complete development quickly. Dybas (1969) found choruses of periodical cicadas in the Chicago metropolitan area in 1969, four years before the emergence of Brood XIII, the only brood in the area; such choruses were also found in 2003 (Cooley et al., 2016). Broods that are offset by 4 years, such as Broods I, V, IX and XIV could be related to each other by the brood derivation scheme Lloyd & Dybas (1966) based on their “4-year acceleration” hypothesis, although as originally published there was no parent–child relationship between Broods I and XIV (Fig. 1). Simon & Lloyd (1982) modified this scheme to include a derivation of Brood I from V and XIV from I via accelerations (Figs. 8A, 8B) to accommodate the Long Island broods (see also Simon, Karban & Lloyd, 1981).

Figure 8 Hypotheses for periodical cicada brood formation based on the original Lloyd & Dybas (1966) “4-year acceleration” hypothesis.

(A) The Simon & Lloyd (1982) modified brood derivation scheme. Solid single arrows indicate that the two broods have contiguous or closely associated geographic ranges, dashed arrows indicate that they do not. Triple arrows indicate that the broods’ ranges are closely associated in three different parts of the country suggesting the possibility of polyphyletic origins (repeated migration between overlapping broods over generations). Broods in parentheses are of doubtful existence consisting of very few records; no arrows lead from them. Broods in square brackets could be derived as shown but there is no need to do so since a simpler derivation exists. We added the parentheses to Brood XV and subtracted them from XI; (B) possible scenario for the formation of disjunct LI broods by four-year accelerations superimposed on the Lloyd and Dybas scheme. Grey broods do not occur on LI. Brood XIV is the most widespread and abundant LI brood. Disjunct LI broods are marked with primes. Doubtful and extinct broods have been removed. (C) Scenario for the formation of disjunct LI broods by three four-year decelerations and one on-year acceleration. (D) Our new four-year jump model for deriving all 17-year periodical cicada broods from a postulated Brood XIV ancestor, by a combination of 4-year and 1-year accelerations, modified from Lloyd & Dybas (1966) and Simon & Lloyd (1982). Broods for which there are no or doubtful historical records are excluded. Double-headed arrows have been added to show both accelerations and decelerations. Triple arrows indicate the possibility of continuous migration through time between geographically overlapping populations over generations. Black arrows indicate jumps that are hypothesized to occur with high probability, grey arrows indicate jumps that are hypothesized to be rare or that do not occur today but may have occurred in the past. Dashed arrows indicate doubtful processes due to lack of geographic proximity. Broods in parentheses are probably not self-perpetuating at present but may have been in the past. Broods in square brackets are extinct. Under this scheme, one-year decelerations are unlikely and are not shown.

Four-year decelerations have likely also played a role in Magicicada evolution. We hypothesize that decelerations occur during colder times when growing seasons are shortened and an extra four years are needed for cicadas to complete development. A large chorusing population of Brood IV emerged four-years late north of Omaha, Nebraska in 2002, where no other cicada broods are found (reported in Marshall, Cooley & Hill, 2011). Similarly, Maier (1985) described reported Brood VI populations in the Hudson River Valley as possible four-year decelerations from resident Brood II populations and also suggested that the miniature broods on Long Island may have arisen from decelerations without offering an explanation for the phenomenon. Aside from such straggling, some brood relationships appear best explainable by four-year delays; a disjunct of Brood I in northeastern Tennessee, 300 km southwest of the main range of the brood (Cooley, 2015), is partially surrounded by Brood XIV and may have been derived by a four-year deceleration from it (Fig. 9). On Long Island, a population synchronous with Brood I appeared repeatedly in exactly the same woods and oviposited in exactly the same trees as Brood XIV that appeared four years earlier (Simon, Karban & Lloyd, 1981). Thirteen-year cicadas also undergo decelerations; Marshall, Cooley & Hill (2011) explained the unexpected emergence of light choruses within the territories of 13-year broods as 4-year late cicadas. Finally, Lloyd & Dybas (1966) suggested that a 4-year deceleration was responsible for the initial formation of 17-year cicadas from 13-year ancestors.

Figure 9 Periodical cicada Brood I disjunct populations in relation to other broods.

Brood I (purple) in the Shenandoah Valley of Virginia with a disjunct population of 200 km southwest in northeast TN that is partly surrounded by Brood XIV (green). Broods II (red), V (orange), VI (blue stars), IX (pink) and X (blue) also shown. Other broods omitted for clarity. Redrawn from (Cooley et al., 2009; Cooley et al., 2011; Cooley, 2015; Cooley et al., 2015).

The eastern section of the main range of Brood V

While the Ohio Valley portion of Brood V occupies a heterogeneous region, the gap between the Shenandoah and Ohio Valley portions of the brood coincides with a shift between two major watersheds; the Shenandoah portion of Brood V lies in the James River drainage adjacent to Brood I, while the Ohio Valley portion occupies the Ohio River drainage adjacent to present day Brood VIII (Figs. 3, 6). Drainage changes may be important in shaping the boundaries of other periodical cicada broods (Cooley et al., 2013; Cooley et al., 2016), and the divide between watersheds may explain the gap between the two major portions of Brood V, the eastern portion in the James River watershed being derived from Brood I by a 4-year deceleration, and the western portion in the Ohio River watershed derived from Brood IX (the supposed precursor of Brood VIII in the Lloyd & Dybas, 1966 scheme) by a 4-year acceleration. These hypotheses can be tested by new data from complete mtDNA genomes and endosymbiont genomes currently under investigation.

The Long Island disjuncts

While some schemes for the derivation of Long Island disjunct “miniature broods” from ancestral Brood XIV populations involve both 1- and 4-year changes (Fig. 8B), a scheme involving only 4-year changes (Fig. 8D) provides a simpler explanation. Four-year jumps alone can explain the Long Island broods via the following deceleration series: [XIV]+4 → [I]+4 → [V]+4 → [IX]. The small Long Island disjunct population of Brood X could then have been derived from a single four-year acceleration from Brood XIV during warmer times [XIV]-4 → [X]. These insights suggest that the four-year acceleration hypothesis should be updated and renamed the “four-year jump hypothesis” to accommodate both life cycle accelerations and decelerations (Fig. 8D).

The eastern PA Brood V disjunct and the four-year jump model

While a 4-year jump model can account for the timing and geography of the Long Island broods, the Carbon County Pennsylvania population of Brood V is not easily accounted for by single 1- and/or 4-year life-cycle jumps from surrounding populations of Brood II or nearby populations of Brood XIV. Instead, at least two 1- and/or 4 year changes are required to derive Brood V from either of these potential parents. In one scenario, the ancestors of Carbon County Brood V were Brood XIV cicadas, which underwent one 4-year deceleration (delay) to join the Brood I schedule, followed by another 4-year deceleration to adopt their current Brood V schedule [XIV]+4 → [I]+4 →[V]. In the second scenario, the ancestors of this Brood V population were Brood II cicadas that underwent a 1-year acceleration to join the Brood I schedule followed by a 4-year deceleration to become Brood V [II]-1 →[I]+4 → [V]. Alternatively, the ancestors could have been Brood II cicadas that underwent a 4-year deceleration (joining the Brood VI schedule) followed by a 1-year acceleration [II]+4 → [VI]-1 → [V]. These scenarios, involving either a Brood XIV or a Brood II ancestor, each require at least one 4-year deceleration. Of these scenarios, a Brood II ancestor seems more plausible because a) no populations of Brood XIV are currently found nearby (Cooley et al., 2011), and b) present-day Carbon County Brood V populations are encircled by Brood II (Cooley et al., 2015). However, all of these hypotheses are diminished by the complete absence of any cicadas on a Brood I or Brood VI cycle in or near Carbon County (Cooley, 2015); thus, each scenario requires that the shift away from the intermediate stage of either scenario be so complete that it left behind no local populations on the intermediate schedule. A similar problem of multiple shifts leaving no intermediates arises in explaining the origin of populations of Brood II in eastern Oklahoma, found occupying a gap within Brood IV (Cooley et al., 2015).

Concluding Remarks

In one sense, the brood concept is a bookkeeping tool; periodical cicada broods are numbered sequentially with an arbitrary start date of 1893 (Marlatt, 1902). In another sense, the brood concept seems to reflect at least some biological reality. To persist, broods rely on sufficient density and geographic spread to effect predator satiation (White & Lloyd, 1979; Lloyd & White, 1980; Karban, 1982a; Karban, 1982b; Williams & Simon, 1995); this reliance on high densities selects for temporal and spatial cohesiveness because small numbers of potential founders are unlikely to persist for long. Yet such cohesiveness is not absolute, or broods would not give rise to other broods. Our study adds to the accumulating evidence that year classes previously thought to have single evolutionary origins are, in a sense, polyphyletic, with multiple origins. In turn, the increasingly evident polyphyly of many broods suggests that periodical cicada life cycles are more plastic than previously thought (Marshall, Cooley & Simon, 2003; Marshall, Cooley & Hill, 2011; Marshall, Hill & Cooley, 2018) and that there may be a heretofore underappreciated tension between life cycle plasticity and selection for strict brood cohesiveness. Even so, successful life cycle switching appears infrequent enough that the broods’ spatiotemporal patterning seems interpretable in light of past climate and landscape changes given sufficient integrative data.

Supplemental Information

Supplemental Information 1 Historical newspaper accounts of periodical cicada emergences in or near Carbon County, Pennsylvania

Click here for additional data file.

Brad Bolton, David Rothenberg and Dan Gilrein, Cornell Cooperative Extension Entomologist Suffolk Co. NY, assisted with field work, as did David Marshall, who also assisted with manuscript preparation. Thanks also to Associate Editor Graham Wallis and two anonymous reviewers for their helpful comments on the submitted version of the manuscript. Full records, many with species information, are available at http://www.magicicada.org. Any opinions, findings, and conclusions or recommendations expressed in this material are those of the authors and do not necessarily reflect the views of the NSF or the University of Connecticut.

Additional Information and Declarations

Competing Interests

Author Contributions

DNA Deposition

Data Availability

The authors declare there are no competing interests.

John R. Cooley conceived and designed the field studies, performed the field studies, analyzed the data, contributed reagents/materials/analysis tools, prepared figures and tables, authored and reviewed drafts of the paper, approved the final draft.

Nidia Arguedas conceived and designed the field studies, performed the field studies, analyzed the data, authored sections and reviewed drafts of the paper, approved the final draft.

Elias Bonaros, Gerry Bunker, Annette DeGiovine, John Knox, Dan Mozgai, Roy Troutman and John Zyla performed the field studies, approved the final draft.

Stephen M. Chiswell performed the field studies, reviewed drafts of the paper, approved the final draft.

Marten Edwards conceived and designed the field studies, performed the field studies, authored sections and reviewed drafts of the paper, approved the final draft.

Diler Haji conceived and designed the field studies, performed the field studies and molecular experiments, analyzed the data, authored sections and reviewed drafts of the paper, approved the final draft.

Hiroki Hasegawa performed the molecular experiments and approved the final draft.

Diane Hassanieh performed the field studies, analyzed the data, approved the final draft.

Gene Kritsky conceived and designed the field studies, performed the field studies, analyzed the data, contributed reagents/materials/analysis tools, authored sections and reviewed drafts of the paper, approved the final draft.

Carolyn Mills performed background research, authored sections and reviewed drafts of the paper, approved the final draft.

Teiji Sota conceived and designed the molecular study, analyzed the data, prepared a figure and a table, authored a section, reviewed drafts of the paper, and approved the final draft.

Jin Yoshimura conceived and designed the field studies, performed the field studies, contributed reagents/materials/analysis tools, approved the final draft.

Chris Simon onceived and designed the field studies, performed the field studies, analyzed the data, contributed reagents/materials/analysis tools, prepared figures and/or tables, authored sections of the paper, reviewed drafts of the paper and approved the final draft.

The following information was supplied regarding the deposition of DNA sequences:

The sequences reported in this paper have been deposited in the DNA Data Bank of Japan Accession Numbers AB740808, AB740809, AB740810, AB740826, AB740827, AB740828.

The following information was supplied regarding data availability:

The raw data are included in the figures.

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
