# Peer review of "The periodical cicada four-year acceleration hypothesis revisited and the polyphyletic nature of Brood V, including an updated crowd-source enhanced map (Hemiptera: Cicadidae: Magicicada)"

_PeerJ, doi:10.7717/peerj.5282_

## Round 0.1 · original submission · Minor Revisions

Although we only have two reviews, there is much common ground in their critiques. First, better background should be given for non-experts on the group. The Figures are quite demanding (I can't see any smaller symbols on Fig2). Reviewer1 makes some apposite suggestions. Second, there is no clear statement of the hypothesis being tested, i.e. at the end of the Intro; the current sentence there is both insufficient and obscure. This makes the manuscript appear to be rather descriptive of distributions, when there is in fact a better underlying story to be told. With respect to the point about nuclear genes, my guess is that they would be uninformative at this level?

I have selected "minor revisions" because there are few technical issues- it is largely about conveying the story more clearly and effectively to the less informed about periodical cicadas (most readers).

Reviewer 1 ·

Basic reporting

This paper presents new information that is one part in a much bigger puzzle about the evolution of periodic cicada. Although each piece has merit on its own, without providing more context it is not possible for a non-expert in periodic cicadas to really understand this current study. Thus the manuscript needs expanding to provide background and detail. But, more importantly, the paper also needs to present clearly stated expectations/predictions of the hypotheses being investigated (see box below called "2. Experimental design").

Please expand to provide background
4. For example in the methods the authors say records of adult cicada emergence were used to map the distribution of Brood V (lines 69-70) and then nine references are provided for studies using the same method, but without a summary of the “method” it is unclear if the procedure involved anything more than that described in the following sentences (find a singing adult cicada, identify it to species level (using morphology or sound?), and write down the location as gps point?).
5. Another example where greater background is required is Figure 6, that presents distribution of haplotypes, but there are no networks or trees to show the relationships of these haplotypes.
6. And the authors need to explain why it would be expected for two distinct cicada species to have the same mitochondrial haplotype lineage! (Lines 141-142).
7. Do 17-year cicada populations change into 13-year cicada populations? Or do 17-4 become 17-year on a new cycle? How would you tell the difference between these two scenarios? Explain what each hypothesis would predict and why you can exclude one or the other.
8. Why no information about host species? 17 years eating plants – the distribution of the host species might matter?

Experimental design

importantly, the paper also needs to present clearly stated expectations/predictions of the hypotheses being investigated.

The hypothesis of parent-offspring relationships between cicada broods is the focus of this work. However, the authors do not say how they could discriminate among hypotheses. After reading the manuscript I felt that detailed distribution maps were not enough to test the existing four-year acceleration hypotheses.
1. Although some mtDNA data is presented it is not clear that the limited diversity is enough to differentiate among hypotheses (no networks shown). If the mtDNA is enough – then present this in the introduction as a prediction figure or as possible alternatives that are expected under different scenarios.
2. Nuclear markers would seem appropriate. What do they tell us?
3. If nuclear markers are not available then perhaps host plant species’ distributions or climate data could be presented? If a large number of individual cicada all switch in the same year (and emerge 4-years early/late) then one would predict the area involved would have a distinct climatic (or host species?) feature not seen outside of the region? I recommend using some simple climate models with (clearly stated) prediction that “watersheds” of interest will be distinctive from areas where the cicada brood didn’t switch.

Validity of the findings

no comment

Additional comments

Minor
Line 31: be explicit that the “puzzle pieces” refer to the distribution map in the US of adult cicada emergence
Line 55: consider rephrasing
Line 58: proximity to (or in contact with) Broods ….
line 80: tautology
Line 124: I recommend not beginning a sentence with “Our map shows”. Here and elsewhere the sentence can start with the observation/result: “The distribution of emergence of adults of Brood V cicadas consists…” and the figure number placed at the end of the sentence. See also line 214
Line 232: Explain “watershed”
Lines 279-298: Too much detail. I sugest you add this information to an annotated figure.

Reviewer 2 ·

Basic reporting

No comment

Experimental design

No comment

Validity of the findings

no comment

Additional comments

This is a nicely researched, comprehensive study of one of the major 'broods' of Magicicada showing a detailed distribution with focus on the particularly interesting areas of parapatry with other broods. The study also shows the good use of crowd sourced data that are important for the comprehensive mapping effort. The text could be slightly more clear at the beginning pointing out the presumption of in situ (anagenetic) change versus the movement of populations with fixed life cycles into new areas, which could also result in the apparent switch, but apparently is not considered. It would also be good to comment on how these broods exclude each other physically. For example, why could you not have a mass emergence in an area every, say, 5 years? I think most readers not familiar with this system would appreciate some basics about the general thinking of cicada research. The life cycle switching hypotheses at the end of the paper are entirely descriptive and hypothetical, so I wonder what is gained from them. There is no avenue shown for how they may be tested. The paper says that the study will provide "key insights into the tension between life cycle plasticity and selection for strict brood cohesiveness" but this seems a rather empty statement based on the data generated here and the discussion of these data. What exactly can we learn about plasticity and selection? Perhaps these mapping studies are simply intriguing because of the curious life cycle phenomenon and the awareness of these cicadas in the general public, but if there is such general evolutionary or ecological question to be addressed based on the mapping exercises it may be interesting to be more specific.

---

## Round 0.2 · accepted · Accept

This ms is a better rounded presentation of the work- thank you for taking on board the reviewers' recommendations.

I have carefully read the entire ms, and it looks good, therefore there is no need to return it to the prior reviewers.

#